# Local Style Tokens: Fine-Grained Prosodic Representations For TTS Expressive Control

*Martin Lenglet, Olivier Perrotin, Gérard Bailly*

Univ. Grenoble Alpes, CNRS, Grenoble INP, GIPSA-lab, France

{martin.lenglet,olivier.perrotin,gerard.bailly}@grenoble-inp.fr

## Abstract

Neural Text-To-Speech (TTS) models achieve great performances regarding naturalness, but modeling expressivity remains an ongoing challenge. Some success was found through implicit approaches like Global Style Tokens (GST), but these methods model speech style at utterance-level. In this paper, we propose to add an auxiliary module called Local Style Tokens (LST) in the encoder-decoder pipeline to model local variations in prosody. This module can implement various scales of representations; we chose Word-level and Phoneme-level prosodic representations to assess the capabilities of the proposed module to better model sub-utterance style variations. Objective evaluation of the synthetic speech shows that LST modules better capture prosodic variations on 12 common styles compared to a GST baseline. These results were validated by participants during listening tests.

**Index Terms**: speech synthesis, expressive TTS, style control, prosody modeling

## 1. Introduction

Latest neural Text-to-speech models (TTS) [1, 2], combined with neural vocoders [3, 4] achieve high standards in terms of naturalness. However, these systems still struggle to model the variability of expressive speech. Two main factors are pointed out to explain these difficulties: 1) the lack of labelled data and 2) the design choice of architecture which enables to learn this variability, as well as its control at inference time. One of the most successful model to tackle both issues is the Global Style Token (GST) architecture [5]. The GST design relies on an reference encoder [6], which converts a reference audio sample into a fixed-size vector which summarizes paralinguistic information. A set of unconstrained tokens are simultaneously trained as an attempt to disentangle main speech features within this paralinguistic representation. Although this architecture enables training on data that is not expressive-labeled, unconstrained tokens are hard to interpret, and post-hoc analysis is necessary to efficiently control the desired synthesis style [7].

Later studies elaborated on improving the expressive control provided by such architectures. Through supervised training [8] or automatic exploration of latent spaces [9], these progress have enabled the careful design of the utterance-wise style bias to be applied in order to generate speech following the target style, without the need for an explicit audio reference. However, natural expressive speech relies on multiple levels of variations. The prosodic structure of one's speech not only depends on one's intents or style, but also on the content itself, as syntactic and semantic structures play an important role in the organization of stress and phrasing [10, 11]. As a result,

utterance-wise style embeddings may lack finer-grained representations in order to fully mimic natural voice behavior.

In this paper, we propose to model fine-grained prosodic patterns through an auxiliary module called ***Local Style Tokens (LST)***. Extending the GST implementation on a segmental level, this module learns to model the residual local speech variations that remain to be explained after utterance-style bias is applied. The proposed module can be applied at multiple scales, providing that such scale can be automatically inferred from the textual input. In this paper, this module was evaluated on Word-level and Phone-level. After discussing related works in Section 2, Section 3 describes the LST specificities and implementation. Objective evaluations described in Section 4.2 compare this module's performances with the GST utterance-wise control and the natural speech. Finally, Section 4.3 describes the listening test procedure we conducted and its results.

## 2. Related Work

Fine-grained prosodic representations have been proposed in TTS before. By construction, pitch and energy embeddings in FastSpeech2 variance adaptor [2] are spectrogram frame-level prosodic embeddings. These provide some prosodic control at inference, but also helps better modeling fundamental frequency. The LST module relies on the same mechanism as prosodic predictors, by re-injecting prosodic representations within the model in the layer they are predicted from.

More focused toward expressive control, [12] proposed to enhance Tacotron2 [1] with word-level style embeddings that are concatenated to the encoder output. Word-level representations are computed with recurrent layers, and then passed to a style attention layer similar to GST [5]. This work inspired us for the present study, but we tried to avoid its main limitation: authors had to train a Prior Encoder, which predicts word style embeddings from the text input in order to synthesize text without audio reference. As a result, the output synthesis is solely based on the text input, denying the choice of expressive style at inference. On the contrary, we aim to use Word (or Phone)-level information to locally refine an global utterance-wise style bias, and therefore combine both style and content inputs.

Hierarchical TTS models like CHiVE [13] or MSE-moTTS [14] also take advantage of the multi-level aspect of speech, by combining intermediate representations from different scales: phonemes, syllables, words, utterance, etc. The entire architectures of these models are built on this hierarchical representations. On the other hand, the proposed LST module is independent; it can be plugged to any encoder-decoder TTS architecture, with various scopes of representation.

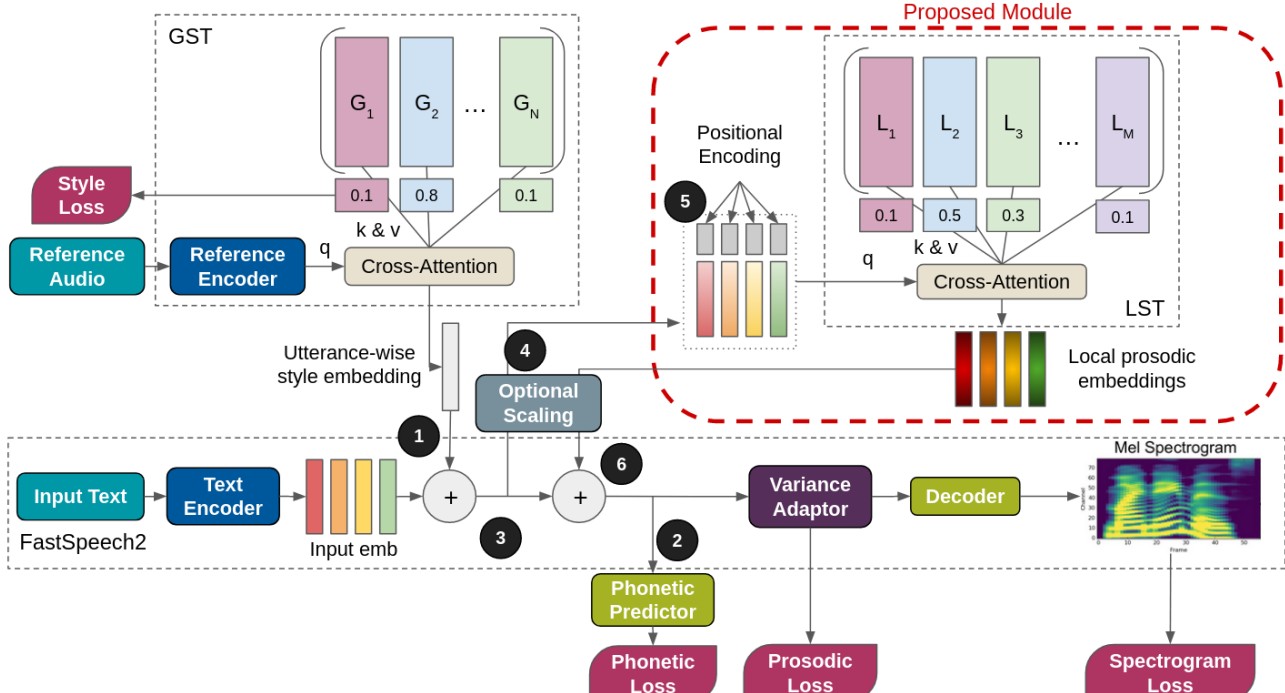

Figure 1: *Model Architecture. The Local Style Tokens module (LST) is plugged after the addition of the utterance-wise style bias.*

# 3. Proposed Model

This section describes the architecture of the proposed Local Style Tokens (LST) module and how it is integrated in the GST-enhanced FastSpeech2 pipeline. The overall architecture of the proposed model is shown in Fig.1.

## 3.1. Model Architecture

### 3.1.1. Model Backbone

The backbone of the model is FastSpeech2 [2], whose encoder, variance adaptor and decoder are kept unchanged[1]. In addition, a label-constrained GST module [8] is plugged at the output of the text encoder (Fig 1.1)[2]. This constrained-GST module converts a reference audio sample into a fixed-size vector through a reference encoder [6]. This fixed-size vector is then used as the query of the cross-attention mechanism in an emotion token layer. Similarly to GST [5], this emotion token layer computes weights that measure the similarity between the reference vector and each global style token. Following [8], a cross-entropy loss is added to enforce each token to encode one particular style. The weighted sum of tokens is then added to all phoneme embeddings computed by the text encoder. Contrary to a set of learnable style embeddings, this module helps training the model on heterogeneous style samples. When given the same style as target, one speaker may produce highly variable utterances, with varying intensity of the given style. The constrained-GST module may account for the intensity by using a mixture of tokens for low intensity utterances, even though their label is the same as unambiguous utterances.

Following [15], a phonetic prediction layer is also added at the output of the text encoder (Fig 1.2). This layer predicts a one-to-one mapping between orthographic inputs and phonetic

outputs. The goals of this layer are twofold: first, it helps disambiguating homographs as shown in [16]. Second, it enables to train the text encoder on <orthography|phonetic> pairs without the need for corresponding audio. This eases the training of models out of audiobooks corpora, e.g. through the use of dictionaries.

### 3.1.2. Local Style Token Module

The Local Style Tokens architecture (LST) is introduced as an auxiliary module which further modulates the output of the text encoder. Although this module does not need to be combined with the GST module, the LST layer alone does not provide explicit control of the synthesis style at inference, which is why the constrained-GST is used in this model.

In FastSpeech2 [2], the variance adaptor implements three prosodic predictors which predict duration, pitch and energy from the output of the text encoder. The prosodic losses associated to these predictors constrain the latent space to encode at least representations of these three prosodic features. Similarly, style embeddings [5, 17] that are added to all phoneme embeddings suppose that additional acoustic and prosodic features are at least partially encoded in this latent space. The LST module may be seen as a residual layer which modulates the latent representations that have been uniformly biased by the GST embedding, according to the content or the position of linguistic units in the utterance. This modulation further improves acoustic and prosodic representations in this latent space.

The LST layer follows the same architecture as the original GST [5]. Two levels of local tokens are examined in this study: Word-level and Phone-Level. In the case of Phone-level tokens, this module takes as inputs the globally biased phoneme embeddings sequence (Fig 1.3). For Word-level, this sequence is averaged by word, to compute word-level representations (Fig 1.4). Because our dataset preserves word boundaries and punctuation marks in case of phonetic inputs, pseudo-word representations

---

[1] https://github.com/ming024/FastSpeech2
[2] GST implementation based on https://github.com/taneliang/gst-tacotron2

are also computed for spaces and punctuation marks (or both when consecutive), also by averaging embeddings.

This input is enhanced by a 32-dimensional positional embedding [18], which is concatenated (Fig 1.5). Indeed, similarly to GST [5], the cross-attention mechanism in the LST layer uses dot product attention, which cannot infer relative positions of representations in the input sequence, in opposition with recurrent networks. However, acoustic patterns relative to style generation depends on the syntactic structure of the utterance and on the relative position of units in the utterance. Although such positional encoding has already been added to phoneme embeddings in the text encoder, preliminary studies showed benefits of explicitly enhancing representations with positional encoding.

This input tensor serves as a set of queries for the cross-attention mechanism in the LST layer. A set of weights is computed for each element in the sequence, and the weighted sum of token values constitutes the local prosodic embeddings sequence which is added to the globally biased phoneme embeddings before the variance adaptor (Fig 1.6). In case of Word-level LST, the local prosodic embedding is first duplicated to be added to all phonemes in the given word (resp. pseudo-word). For ease of interpretability of local token weights, the cross-attention mechanism is single-headed.

### 3.2. Training and Inference Processes

During training, the reference mel-spectrogram matches the target output. The reference encoder and the cross-attention GST work as an emotion recognition module which computes a probability distribution of the given audio input on all constrained style tokens. In contrast, the LST weights are not constrained during training. The LST layer does not require additional loss. It is trained by the back-propagation of the spectrogram loss, prosodic predictors losses and phonetic loss. The back-propagation is not stopped at the input of the LST module, which enables the text encoder to incorporate features that may be used to compute local prosodic embeddings in the LST layer. The entire model can be trained simultaneously, from scratch.

Similarly to constrained-GST, two style control methods are available at inference: 1) use a target reference audio which produces a mixture of global style tokens or 2) specify the mixture of global style tokens to use. Because the GST module is constrained, each global style token has been trained to produce one particular style. Thus, one-hot vectors are particularly fitted to generate the desired style. Local prosodic embeddings are computed in parallel by the LST module, which does not impact the inference speed of the model.

## 4. Experiments and Results

### 4.1. Models and dataset

Three models are trained for this study: 1) FastSpeech2 with constrained-GST referred as *GST* (the Baseline) ; 2) Baseline enhanced with word-level LST referred as *LST_W*; and 3) Baseline enhanced with phoneme-level LST referred as *LST_P*. *LST_W* offers more context at the input of the LST module, which may result in a more careful choice of representations in the LST layer. On the other hand, word style bias may result in less intra-word modulation.

All models are trained on the same dataset, given in Table 1. This internal French dataset has been uttered by a French professional theater actress. Sentences are taken from the SI-WIS database [19], which is composed of isolated extracts from French Novels and French parliament debates. For expressive

Table 1: *Expressive Dataset. Durations are given in minutes.*

| Style | Train | | Test | |
|---|---|---|---|---|
| | Duration | # Utt | Duration | # Utt |
| Angry | 24.2 | 523 | 1.5 | 32 |
| Comforting | 32.3 | 488 | 1.6 | 27 |
| Committed | 21.1 | 430 | 1.4 | 29 |
| Enthusiastic | 29.5 | 569 | 1.4 | 28 |
| Obvious | 27.0 | 492 | 1.5 | 27 |
| Playful | 19.1 | 465 | 1.5 | 28 |
| Pleading | 34.2 | 605 | 1.9 | 31 |
| Skeptical | 29.8 | 620 | 1.6 | 32 |
| Sorry | 24.2 | 448 | 1.1 | 23 |
| Surprised | 26.8 | 503 | 1.6 | 32 |
| Thoughtful | 43.4 | 450 | 2.1 | 27 |
| Narrative | 287.6 | 6235 | 14.6 | 307 |
| **Total** | **599.2** | **11828** | **31.8** | **633** |

Table 2: *Number of Local Style Tokens used by the model per style.*

| Style | LST_W | | LST_P | |
|---|---|---|---|---|
| | # Tokens | # Exclusive | # Tokens | # Exclusive |
| Angry | 9 | 1 | 8 | 0 |
| Comforting | 8 | 1 | 7 | 0 |
| Committed | 8 | 0 | 11 | 0 |
| Enthusiastic | 6 | 0 | 10 | 0 |
| Obvious | 7 | 0 | 8 | 0 |
| Playful | 9 | 1 | 11 | 0 |
| Pleading | 6 | 0 | 10 | 0 |
| Skeptical | 10 | 0 | 12 | 0 |
| Sorry | 4 | 0 | 8 | 0 |
| Surprised | 8 | 0 | 11 | 0 |
| Thoughtful | 8 | 0 | 12 | 0 |
| Narrative | 11 | 3 | 13 | 2 |
| **Overall (/32)** | **32** | **6** | **30** | **2** |

speech recording, she was asked to utter the given sentences with the specified style during exercise-in-style sessions. During these sessions, the actress was prompted to start her utterance with a context sentence relative to the style being produced: "I am begging you" for "Pleading", "I do not believe it" for "Skeptical", "Really?" for "Surprised", etc. This context sentence was cut from the final recording. The recordings are being evaluated to verify that the produced style is correctly recognized by naive speakers, but this evaluation is still on-going at the time of writing of this study.

The content is decorrelated from the expressed style, and sentences differ between styles. Sentences that were not uttered with a specific style were labeled as "Narrative". This audio-visual expressive dataset was recorded in the GIPSA-Lab, as part of the Theradia project [20]. The 12 styles were chosen to cover the expressive needs of the Theradia application. Only the audio was used in this study. 5% of the corpus was randomly selected as the test set. All models are trained for 250 epochs using both orthographic and phonetic input representations. Following early implementations of FastSpeech2, the pitch predictor is trained on raw pitch values in semitones, instead of continuous wavelet transforms [21] in latter works. Pitch and energy values are averaged by phonemes, and normalized. The one-to-one phonetic targets for the phonetic prediction task are established using patterns described in [15]. The vocoder used is Waveglow [3].

Table 3: *Mean errors per style computed on the test set. Blue (resp. red) indicates a lower error (resp. higher error) than* **GST**. *\* and \*\* indicate that the distribution statistically differs from the* **GST** *baseline with p<0.05 and p<0.01, respectively.*

| Style | Spectral Error (dB) | | | Duration Error (ms) | | | Pitch Error (Semitones) | | | Energy Error (dB) | | |
|---|---|---|---|---|---|---|---|---|---|---|---|---|
| | GST | LST$_W$ | LST$_P$ | GST | LST$_W$ | LST$_P$ | GST | LST$_W$ | LST$_P$ | GST | LST$_W$ | LST$_P$ |
| Angry | 0.93 | 0.91 | 0.93 | 9.18 | 9.01 | 9.13 | 2.29 | 2.14 | 2.50 | 3.24 | 3.15 | 3.34 |
| Comforting | 0.88 | 0.88 | 0.89 | 11.41 | 11.64 | 11.12 | 1.59 | 1.62 | 1.77 | 3.12 | 3.24 | 3.11 |
| Committed | 1.00 | 0.99 | 0.96 | 9.83 | 9.83 | 9.33 | 4.10 | 4.15 | 3.93 | 3.26 | 3.16 | 3.24 |
| Enthusiastic | 1.18 | 1.16 | 1.18 | 9.82 | 9.45 | 9.82 | 4.31 | 3.93 | 4.31 | 3.03 | 3.10 | 3.06 |
| Obvious | 1.07 | 1.05 | 1.08 | 10.74 | 10.23 | 10.01 | 3.32 | 3.12 | 3.41 | 3.12 | 3.09 | 3.12 |
| Playful | 0.97 | 0.99 | 0.96 | 12.08 | 11.29 | 11.25 | 4.06 | 3.98 | 4.11 | 3.09 | 3.06 | 3.04 |
| Pleading | 0.92 | 0.92 | 0.91 | 9.67 | 9.03 | 9.49 | 1.93 | 1.78 | 1.72 | 2.49 | 2.44 | 2.45 |
| Skeptical | 0.98 | 0.97 | 0.99 | 10.10 | 9.85 | 10.13 | 2.86 | 3.03 | 3.08 | 3.06 | 3.03 | **3.23\*\*** |
| Sorry | 0.67 | 0.68 | 0.67 | 9.50 | 9.50 | 9.70 | 1.05 | 1.16 | 1.04 | 2.64 | 2.75 | 2.77 |
| Surprised | 0.97 | 0.97 | 0.97 | 10.20 | 9.85 | 10.07 | 3.47 | 3.33 | 3.40 | 3.21 | 3.13 | 3.30 |
| Thoughtful | 0.94 | 0.95 | 0.97 | 22.23 | 22.28 | 22.52 | 2.51 | 2.63 | 2.52 | 2.86 | 2.91 | 2.96 |
| Narrative | 0.90 | 0.90 | 0.89 | 10.45 | **10.36\*** | 10.53 | 2.75 | 2.73 | 2.74 | 2.93 | **2.86\*** | **2.86\*\*** |
| **Total** | 0.93 | 0.93 | 0.92 | 10.52 | 10.31 | 10.44 | 2.70 | 2.67 | 2.71 | 2.97 | 2.94 | 2.96 |

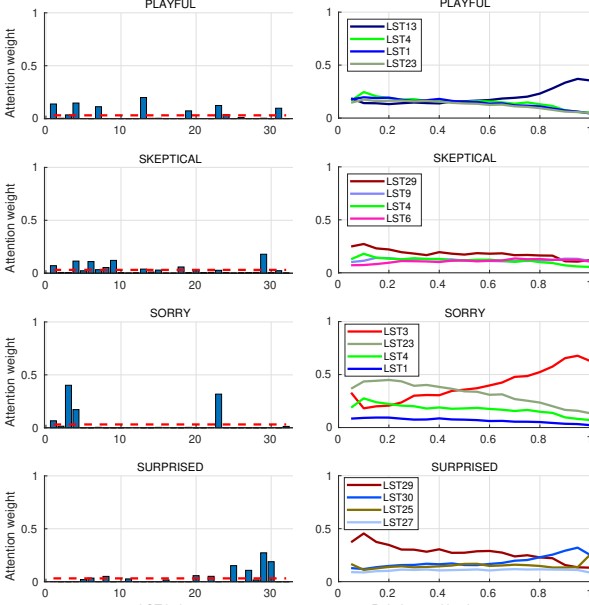

(a) *Mean LST usage by style.*

(b) *Mean LST usage relative to the position in utterance (0: first character, 1: last character).*

Figure 2: *Local Style Tokens usage by style for* **LST$_W$**. *Four styles are shown as examples: "Playful", "Skeptical", "Sorry" and "Surprised". Only the 4 local tokens with the maximum mean attention weights are shown in Fig 2b.*

Following the constrained-GST architecture given by [8], 12 tokens are needed in the **GST** layer to account for each style label cited above. The target styles given to the actress are used as style labels. The number of local style tokens is fixed to 32 for both **LST$_W$** and **LST$_P$**. 32 tokens is chosen as a middle ground between sharing tokens across GST representations and providing enough local tokens so that each global style can rely on dedicated local tokens. To evaluate the usage of each individual local token, 100 utterances of the test set were randomly selected, and each utterance was generated with the 12 styles of the corpus. Mean attention weights of each local token were computed per style. Examples of mean attention weights by lo-

cal token and the dynamic of such attention are given in Fig 2 for **LST$_W$**. Table 2 summarizes the number of local tokens used per style, as well as the number of tokens that are exclusive to the specified style. One local token is counted as used if its mean activation weight is above the uniform distribution across all local tokens (above the red dashed line in Fig 2a). The overall number of tokens used differs from the sum because some tokens are shared across styles. Two tokens are never used by **LST$_P$**. **LST$_W$** and **LST$_P$** with 64 tokens were tested but showed that too many tokens were never used.

The diversity of local tokens usage illustrates the benefits of modelling prosody at a smaller scale. Multiple local tokens are used by all styles to model various local patterns. "Angry", "Comforting", "Playful" and "Narrative" use exclusive local tokens in **LST$_W$**, assessing for unique speech behaviors in this sub-corpus (same for "Narrative" in **LST$_P$**). Figure 2b shows the dynamic of local tokens attention relative to the position in the utterance. Global styles exhibit various patterns, but most characteristic behaviors are found at the beginning (LST29 for "Surprised") and at the end of utterances (LST13 for "Playful" and LST3 for "Sorry"). Other styles like "Skeptical" are more stable, but smaller variations of local tokens usage also indicate that the LST module helps modulating representations at a finer-grain.

### 4.2. Objective Evaluation

Objective evaluations of the synthetic models were conducted to assess the benefits of the proposed model compared to the baseline. Models are evaluated on 3 aspects: training loss criteria, pitch variations and phrasing behaviors. All statistical differences between distributions are evaluated pair-wise through non-parametric Wilcoxon rank sum tests. The objective metrics shown in this section focus on various evaluations of the three main prosodic features: duration, pitch and energy. Other acoustic features like voice quality may impact style modelling [22], but were not measured in this study.

#### 4.2.1. Test Set Errors

All models are trained under the same loss criteria, which include mel-spectrogram losses and prosodic features predictions (duration, pitch and energy). Table 3 summarizes these errors from the test set ground truth (**GT**) after training. Spectral error

Table 4: *Mean standard deviation of pitch per style (Semitones). * indicates that the distribution statistically differs from the GT (p<0.05). Blue (resp. red) indicates that the proposed model performs better (resp. worse) than the GST baseline.*

| Style | GT | GST | LST$_W$ | LST$_P$ |
|---|---|---|---|---|
| Angry | 3.59 | **2.95** | 2.85* | 2.77* |
| Comforting | 1.94 | 1.68 | 1.70 | 1.77 |
| Committed | 3.92 | 3.92 | 3.94 | 3.69 |
| Enthusiastic | 4.79 | **3.27*** | **3.44*** | **3.24*** |
| Obvious | 4.25 | 3.06* | 3.41* | 3.39* |
| Playful | 5.27 | **4.15*** | 4.03* | 4.04* |
| Pleading | 2.90 | 2.36 | 2.45 | 2.51 |
| Skeptical | 4.49 | **2.90*** | 3.36* | 3.04* |
| Sorry | 1.85 | **1.49*** | 1.65 | 1.58* |
| Surprised | 5.66 | **4.03*** | 4.20* | 4.08* |
| Thoughtful | 2.70 | 2.53 | 2.64 | 2.54 |
| Narrative | 4.94 | **3.89*** | 3.91* | 3.89* |

Table 5: *Mean proportion of silences in synthetic vs. GT utterances (in %). ** indicates that distributions statistically differ from the GT with p<0.01. Blue (resp. red) indicates that the proposed model performs better (resp. worse) than the GST baseline.*

| Style | GT | GST | LST$_W$ | LST$_P$ |
|---|---|---|---|---|
| Angry | 2.6 | **2.0**** | 1.8** | 3.0 |
| Comforting | 2.5 | 2.3 | 2.2 | 1.9* |
| Committed | 4.8 | 3.3 | 3.4 | 3.3 |
| Enthusiastic | 1.6 | 1.7 | 1.6 | 1.2 |
| Obvious | 0.8 | 1.1 | 0.8 | 1.0 |
| Playful | 6.6 | 4.7 | 4.8 | 5.2 |
| Pleading | 1.3 | **0.6**** | 0.6** | 0.7** |
| Skeptical | 2.4 | **1.8**** | 2.0** | 1.5** |
| Sorry | 2.2 | **1.4**** | 3.2** | 1.9** |
| Surprised | 2.0 | 1.9 | 1.4 | 1.0 |
| Thoughtful | 1.8 | 2.4 | 1.4 | 1.6 |
| Narrative | 3.8 | **2.5**** | 2.6** | 2.6** |

is computed on synthesis aligned with Dynamic Time Warping (DTW) [23]. Mean euclidean distances are evaluated on the alignment path. Duration and energy errors are computed on all phonemes, while pitch error is only evaluated on vowels.

Lower errors indicate that models that implement the LST modules produce speech closer to the GT for most styles. Over all errors, LST$_W$ provides the most consistent benefits, with 28 improvements and 14 degradations, compared to 20 improvements and 20 degradations for LST$_P$. These improvements were significant for "Narrative", but not for the other styles. "Committed", "Enthusiastic", "Pleading", "Surprised" and "Narrative" are the most improved styles. This indicates that those five styles rely on local prosodic patterns that are difficult to model with utterance-wise style representation. On the other hand, "Comforting", "Skeptical", "Sorry" and "Thoughtful" show higher errors with LST. Overall, the more mitigated results of LST$_P$ may be explained by the wider variability provided by local tokens at the phoneme scale. This variability opens the door for more risks of divergence with GT.

While lower errors indicate that synthetic speech is closer to the natural utterances recorded in our corpus, there is no golden standard for conveying a given style. Many variants: 1) could have been performed by the recorded speaker for this same sentence and style, and 2) may be perceived as similarly expressive for a human listener. As a result, the GT is not the only licit speech production, and more objective evaluations are needed to assess the expressive quality of the synthetic speech. In the following, we then compare distributions of prosodic parameters measured on GT and on each of our models. Our criteria for a successful rendering of prosodic features is therefore to have **non**-significant differences between a model and the GT.

### 4.2.2. Pitch standard deviation

Pitch standard variations by utterance is commonly used to evaluated expressive capabilities of TTS models [2, 12]. Table 4 compares the pitch variability of GT to that of the synthetic models. Highly variable styles like "Enthusiastic", "Obvious", "Playful", "Skeptical" and "Surprised" are harder to model for TTS, as shown by statistical differences between GT and all synthetic models. Overall, the LST module helps generating more pitch variability, even though results were not significant. Significant improvements were found for "Sorry", with LST$_W$ generating pitch standard deviations closer to GT.

### 4.2.3. Phrasing Error

Phrasing is decisive in perceptual judgements [24, 25]. Notably, varying frequency of silences when modifying the speaking rate is a key feature of natural voice that synthetic models generally struggle to achieve. Table 5 shows mean silence proportions per style for each model and GT. Significant differences between GT and synthetic models for "Pleading", "Skeptical", "Sorry", and "Narrative" demonstrate the difficulties of TTS to replicate natural balance between speech and silences for these styles. The LST module does not provide much improvement in that regard. Conversely, LST$_P$ produces more pauses for styles with high silences ratio like "Angry" and "Playful", whose natural behaviors are hardly replicated by utterance-wise style bias in GST (this improvement was significant for "Angry").

Duration modulation were also evaluated as an indicator of local prosodic patterns. We hypothesize that polysyllabic words should be more impacted by local modulations, as they are mostly content words. At least some of the studied styles should emphasize local key points in the utterances that are embodied by content words. Word duration modulation is evaluated as the ratio between the duration of the last vowel and the mean duration of other vowels of the same word. This measure indicates the lengthening of last syllable of polysyllabic words, as approximation of content words. Table 6 summarizes evaluated duration modulation per style. Lengthening of the last syllable of polysyllabic words is very common in GT, as shown by mean word duration modulations above 1.25 for every style. "Obvious", "Pleading", "Sorry" and "Thoughtful" show the higher degree of modulation. This modulation is closely replicated by all synthetic models, with slight variations between models. Interestingly, GST tends to elongate durations excessively, in particular on "Enthusiastic", "Playful" and "Thoughtful", while the LST modules help producing more natural duration modulations.

Table 6: *End syllable duration modulation evaluated on poly-syllabic words. \* indicates that the distribution statistically differs from the GT (p<0.05). Blue (resp. red) indicates that the proposed model performs better (resp. worse) than the GST baseline.*

| Style | $GT$ | $GST$ | $LST_W$ | $LST_P$ |
|---|---|---|---|---|
| Angry | 1.34 | 1.34 | 1.28 | 1.34 |
| Comforting | 1.33 | 1.35 | 1.39 | 1.36 |
| Committed | 1.34 | 1.34 | 1.39 | 1.34 |
| Enthusiastic | 1.36 | 1.47 | 1.44 | 1.38 |
| Obvious | 1.41 | 1.43 | 1.41 | 1.40 |
| Playful | 1.32 | 1.54 | 1.44 | 1.51 |
| Pleading | 1.37 | 1.30 | 1.35 | 1.30 |
| Skeptical | 1.24 | 1.28 | 1.31 | 1.22 |
| Sorry | 1.43 | 1.44 | 1.52 | 1.46 |
| Surprised | 1.25 | 1.23 | 1.27 | 1.21 |
| Thoughtful | 1.88 | 1.95 | 1.91 | 1.95 |
| Narrative | 1.33 | **1.36\*** | **1.36\*** | **1.37\*** |

Table 7: *Expressive-MUSHRA results per style. Blue (resp. red) indicates that the proposed model performs better (resp. worse) than the GST baseline. \* and \*\* indicates that this difference with GST is statistically significant with p<0.05 and p<0.01, respectively. LA = Low Anchor, GT = Ground-Truth.*

| Style | LA | $GST$ | $LST_W$ | $LST_P$ | $GT$ |
|---|---|---|---|---|---|
| Angry | 17.3 | 63.0 | 64.3 | **68.3\*\*** | 75.6 |
| Comforting | 15.4 | 66.2 | 63.5 | 61.5 | 80.5 |
| Committed | 24.9 | 65.1 | **70.9\*\*** | 68 | 76.4 |
| Enthusiastic | 11.6 | 66.2 | 70.0 | **74.0\*** | 86.4 |
| Obvious | 40.2 | 65.7 | 61.4 | 65.3 | 84.7 |
| Playful | 16.4 | 63.3 | 66.3 | 67.4 | 86.5 |
| Pleading | 12.3 | 71.3 | 70.1 | 71.2 | 77.9 |
| Skeptical | 36.3 | 47.3 | 50.6 | 46.6 | 63.3 |
| Sorry | 15.4 | 63.2 | **71.1\*\*** | 68.0 | 68.7 |
| Surprised | 14.3 | 78.5 | 75.6 | 73.7 | 85.3 |
| Thoughtful | 24.3 | 46.9 | 47.5 | 52.7 | 62.7 |
| Narrative | 64.6 | 63.1 | **67.4\*** | **67.5\*** | 69.5 |
| **Total** | 24.2 | 63.0 | 64.7 | 65.0 | 76.1 |

### 4.3. Listening Experiment

In order to evaluate perceptual differences between the proposed model and the baseline, 60 participants took part in an online MUSHRA-like experiment [26], run with the framework webMUSHRA [27]. Given the text uttered and the target style, participants were asked to evaluate on a scale from 0 (very bad) to 100 (excellent) if the style was correctly rendered. For this listening test, we selected 10 utterances per style that maximize spectral distances between systems (120 in total). 5 groups of 12 participants evaluated each 24 utterances (2 per style), with 5 systems per utterance: the *GST*-enhanced FastSpeech2 baseline, the two proposed models $LST_W$ and $LST_P$, the vocoded *GT* (high anchor), and a FastSpeech2 without GST trained on non-expressive data (low anchor) referred as *LA*. Because the Ground-Truth is not the only way to convey the given style, it was not given as an explicit reference to the participants during the listening test. Participants who misunderstood the evaluation task were excluded: it includes ranking the non-expressive model higher than the other models, as well as participants with significantly lower standard deviation of grades. Examples rated by participants can be found at the following link[3].

Results of this perceptual experiments are given in Table 7. *LA* was ranked significantly lower than all other models, except for "Narrative" which is also modelled by the non-expressive *LA*. Participants tend to favor $LST_W$ and $LST_P$ over *GST*. Most noticeable improvements are found for "Angry", "Committed", "Enthusiastic", "Sorry" and "Narrative". Objective evaluations have shown that the LST module helps producing local behaviors that are closer to the *GT*. Reproducing pitch variations and phrasing is critical for these styles to be perceived as natural. Note that *GT* exhibited relatively poor results on "Skeptical" and "Thoughtful". These styles may have been too caricatured by the speaker, which participants judged as unnatural.

---

[3] `https://www.gipsa-lab.grenoble-inp.fr/~martin.lenglet/listening_page_LST/index.html`

## 5. Conclusions and Discussion

In this paper, we proposed the LST module for expressive TTS which helps modeling fine-grained prosodic patterns. This module was evaluated on 12 common expressive style for French synthesis. Most promising improvements over the GST baseline are shown for "Angry", "Committed", "Enthusiastic" and "Sorry", for which more subtle prosodic variations are needed to achieve a natural behavior.

The number of tokens and training process of the LST module deserves more attention. The best results are found for styles that make use of multiple local tokens (Table 2 and Fig 2). This result was expected, since adding the same local token all along the utterance should not provide different results from an utterance-wise style bias. Constraining the LST module to maximize tokens usage should help the model showing more robust results. Additionally, the number of local tokens should be adapted to the scale of representations, e.g. allowing more various contributions for finer-grained prosodic patterns. Finding the acoustic and prosodic features encoded by the local tokens may also help understanding the acoustic similarities between styles. This analysis is left for future works.

This study reinforces the need for more elaborated evaluation paradigms for expressive speech. While style "Sorry" showed the greater amount of objective errors compared to the Ground-Truth, it was still perceived as well rendered during listening tests. Prosodic patterns followed by the Ground-Truth are not exclusive, and evaluation has to be adapted to match perceptual judgements.

We will explore cascaded LST that can be stacked to encode increasingly finer representations such as phrases, words, syllables, phonemes, etc. It would also be interesting to explore the addition of level-specific information, using pre-trained representations as BERT [28] for example.

## 6. Acknowledgements

This research has received funding from the BPI project THERADIA and MIAI@Grenoble-Alpes (ANR-19-P3IA-0003). This work was granted access to HPC/IDRIS under the allocation 2023-AD011011542R2 made by GENCI.

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
