# OpenReview forum: "Local Style Tokens: Fine-Grained Prosodic Representations For TTS Expressive Control"
_Interspeech.org/2023/Workshop/SSW — SSW12_

### Official Review · Reviewer_Cg8s · 2023-05-30
**Local Style Tokens: should be accepted**

**Rating:** 7
**Confidence:** 4

**Review:**

Key Strength of the paper: Generally well-written, technically sound, includes both objective and subjective evaluation, good summary of related work.

Main Weakness of the paper: Lack of motivation for certain design decisions. (1) How was the set of 12 styles established; how are they defined? (2) How was it established that the actress indeed produced the target style? (3) The classical acoustic-prosodic features (pitch, duration, intensity) are modeled, but expressive speech is strongly conveyed by changes in voice quality, which supposedly contributes to convey (some of) the 12 styles, too. No attempt was made to model VQ, nor was it even mentioned.

Novelty/Originality, taking into account the relevance of the work for the SSW audience: Highly relevant.

Technical Correctness. The work appears to be technically solid. Some details are lacking that would allow a reproduction of the modeling and experiment, primarily details about the definition of expressive styles and the elicitation and validation procedure.

Suggestions for improvement: see above.

Quality of References: Generally adequate, but Liberman/Prince and Selkirk are not exactly the ideal references for expressive speech.

Clarity of Presentation: Generally clear (but see under weaknesses). The paper would benefit from proofreading for spurious errors (perceptive -> perceptual, vary low -> very low, etc., as well as a few grammatical errors).

---

> ### Author Response · Authors · 2023-06-28
> **Description of the expressive dataset**
>
> Thank you for your comment. You are right about the lack of description of the dataset. We were not able to provide an explicit description of our corpus for blind reviews, but some additional information were added in the final revision. However, the recording of the dataset is recent, so additional evaluations have to be performed to verify that the target styles have been produced correctly. The results of this evaluation are not published at the time of writing of this study. For now, we have taken the target style as labels for the training. Voice quality was not evaluated for this study, but it may indeed convey expressive styles, so it has been added as a remark and left for future works.

---

### Official Review · Reviewer_q2MJ · 2023-06-02
**Timely solution to address prosody modelling for Expressive TTS with a thorough analysis**

**Rating:** 9
**Confidence:** 5

**Review:**

***** Key Strength of the paper

- in-depth evaluation and analysis
- timely proposition with a flexible solution
- paper clearly written

***** Main Weakness of the paper

***** Novelty/Originality, taking into account the relevance of the work for the SSW audience

The topic addressed is highly relevant for the workshop, and the proposition is novel in that it addresses an important problem faced by the TTS community in a flexible manner.

***** Technical Correctness

The evaluation and analysis are thorough.

***** Quality of References

The references seem to be adequate and well balanced

***** Clarity of Presentation

The paper is clearly written and easy to follow

***** Suggestions for improvement

These suggestions remain minor
- It would be clearer if the reference of the different elements of the figure follows the same
- Instead of MUSHRA-like, I think the author should explicitly state the difference to MUSHRA (which I suspect is the lack of anchors)
- I think the title/caption of Figure 1 is incorrect as it suggests that the overall model is the LST instead of the module
- The colours scheme is a good idea, but some parts seem to have been missed (e.g., 4.6 for committed in Table 2)

---

> ### Author Response · Authors · 2023-06-28
> **Thank you for your interest in our research**
>
> Thank you for your comments and your interest for our research. We have taken your suggestions into account in our final revision. We have stated clearly that the listening test is called MUSHRA-like because their is no explicit audio reference that the participant should find in the stimuli. However there are two anchors: the low anchor is the non-expressive model, and the high anchor is the vocoded Ground-Truth. The title of Figure 1 was also updated.

---

### Decision · Program_Chairs · 2023-06-14

**Decision:**

Accept

**Comment:**

Dear authors,

SSW2003 received 45 papers. The acceptance rate is 82%. We are pleased to inform you that your paper has been ACCEPTED by the SSW2023 Programme Committee.

Please read the reviews carefully and submit your camera-ready paper by June 28th. Most of reviewers performed a detailed review. Please answer to their questions and take into account their comments. Note that camera-ready papers are credited of one extra page to allow authors to consider reviewers’ suggestions. So max 7 pages in total including figures & refs.

Regards,
The SSW organizing chairs